# *Botryosphaeriaceae* Species Associated with Stem Canker, Shoot Blight and Dieback of *Fraxinus ornus* in Italy

**Alessandra Benigno [1,*], Chiara Aglietti [1], Giovanni Rossetto [2], Carlo Bregant [2], Benedetto Teodoro Linaldeddu [2] and Salvatore Moricca [1]**

[1] Department of Agricultural, Food, Environmental and Forest Sciences and Technologies, University of Florence, Piazzale delle Cascine 28, 50144 Firenze, Italy; chiara.aglietti@unifi.it (C.A.); salvatore.moricca@unifi.it (S.M.)

[2] Dipartimento Territorio e Sistemi Agro-Forestali, Università degli Studi di Padova, Viale dell'Università 16, 35020 Legnaro, Italy; giovanni.rossetto.4@phd.unipd.it (G.R.); carlo.bregant@phd.unipd.it (C.B.); benedetto.linaldeddu@unipd.it (B.T.L.)

* Correspondence: alessandra.benigno@unifi.it

**Abstract:** A severe dieback of flowering ash (*Fraxinus ornus* L.) has been observed in north-central Italy in the last decades. Symptoms include typical sunken, light-brown cankers on the stem and branches; vascular discoloration; tip and shoot dieback; and foliage necroses. The disease was more evident at the beginning of the growing season, and more severe on young regeneration. Six *Botryosphaeriaceae* species were consistently isolated from symptomatic plant tissues: *Botryosphaeria dothidea*, *Diplodia fraxini*, *Diplodia subglobosa*, *Dothiorella iberica*, *Dothiorella omnivora* and *Neofusicoccum parvum*. *B. dothidea* and *D. fraxini* expressed higher aggressiveness and showed a widespread incidence, being the species most frequently associated with cankers; the other four species were less virulent and more erratic, occurring mainly on succulent branch tips and foliage. Isolates were characterized using morphological and molecular approaches (colony/conidial phenotyping and rDNA-ITS genotyping). Phylogenetic analysis provided congruent phylogenies depicting the relationships of the six taxa with the most closely related conspecifics. Pathogenicity tests on 2-year-old seedlings confirmed the higher virulence of *B. dothidea* and *D. fraxini*. Extensive, multi-year field surveys at different sites supported the hypothesis that climatic vagaries, mainly heat, water and drought stresses, impaired tree health and vigor, facilitating infection and pervasive colonization by these *Botryosphaeriaceae* species. Environmental stressors are thus the key factor bringing the six fungal pathogens together in a multitrophic interaction with *F. ornus* in a novel, lethal fashion.

**Keywords:** flowering ash-dieback; cankers; twig blights; leaf necroses; endophytic *Botryosphaeriaceae*; pathogenicity assays; climate change; environmental stress; Mediterranean area

## 1. Introduction

Flowering ash (*Fraxinus ornus* L.), also known as "manna ash", is a small- to medium-sized deciduous tree species native to hilly and mountainous mixed forests of southern Europe and southwestern Asia [1]. This tree has an asymmetrical, hemispherical or flattened crown composed of leaves that are 20–30 cm long and odd-pinnate, arranged in 5–9 leaflets that are obovate and 7–10 cm long. The bark is dark grey and smooth, even in adult trees [1]. Though the timber of this species usually has low economic interest, *F. ornus* stands have been maintained in various parts of Europe as coppices for producing firewood and small tool handles [2]. The scarce appreciation of *F. ornus* timber is mainly due to this species' pioneer habit that facilitates its growth on slopes and remote places, favoring the development of small and poorly shaped trunks with many timber defects. On the contrary, these features give *F. ornus* high ecological importance, which is

substantiated in its use as a primary component of protective forests and for afforestation of degraded sites [3]. In addition, this species assumes economic value in southern Italy where it is planted to produce the manna, a bitter-sweet tasting sap that is exuded by the plant, crystallizing in the air into yellow masses, used as a sweetener, laxative, and digestive aid [1]. Due to the poor quality of its wood, *F. ornus* always received little attention from humans, and this is why no serious threat had been documented for this tree species until the reporting in Italy of the ash dieback disease caused by the ascomycete fungus *Hymenoscyphus fraxineus* [4]. *H. fraxineus* occurrence has been documented in some Italian regions since 2009, but although the fungus is able to infect all the ash species growing there, i.e., common ash (*Fraxinus excelsior*), flowering ash (*Fraxinus ornus*) and narrow-leaved ash (*Fraxinus angustifolia*), serious damage has been reported only on *F. excelsior* [5–7].

A diverse and more complex disease framework has been observed in recent years in *Fraxinus* forests, mostly concentrated in Mediterranean-climate areas characterized by lack of rainfall and prolonged drought. The varying set of symptoms observed on *F. ornus* collectively generates an undescribed syndrome, which is quite dissimilar from the well-defined symptomology caused by *H. fraxineus* [8–10]. Indeed, this new syndrome includes sunken cankers on the stem and branches, sometimes appearing numerous along the stem of young plants; a characteristic wedge-shaped necrotic sector in the cross section of declining branches and stems; leaf and shoot blight; and progressive dieback of the canopy [11,12].

The above-described progressive and extensive dieback, which is severely affecting *F. ornus* stands in some areas of central and northern Italy, has increased over the years, leading to the necessity to ascertain the etiologic agent(s) as well as the possible predisposing factors involved [13,14]. Among the predisposing factors that could have activated the onset of this new decline/dieback syndrome, climate change seems to have a central role. Climate warming, in fact, by altering temperature and precipitation regimes at a regional scale, can strongly impact plant growth and physiology on the one hand, and modify the life-history strategies and behavior of associated microorganisms on the other. Regional climate anomalies can thus cause profound changes in natural environments, reducing the resilience of forest formations, increasing tree vulnerability and triggering the onset of new diseases [15,16]. The general climate warming trend is particularly exacerbated in the Mediterranean basin, considered one of areas of the planet most threatened by it. In fact, the Mediterranean zone is expected to undergo a higher increase in temperatures than the rest of the planet combined with prolonged drought events followed by extreme weather conditions [17–20]. In this alarming scenario, the ecology, biogeography and infection strategies of plant pathogens can be altered, creating conditions conducive to new disease emergence and spread [15,16,21].

Among the pathogens that can take advantage of climate change are some prominent members of the *Botryosphaeriaceae*. The *Botryosphaeriaceae* family includes more than 200 species that can affect thousands of plant species worldwide [22]. Many of these fungi are acquiring importance as emerging pathogens due to frequent reports of their relationship with dieback syndromes in forest and agro-forest ecosystems [23]. The endophytic lifestyle of these fungi has long been demonstrated, and the main factors linked to the development of diseases caused by these fungi have been identified in stressful environmental conditions [24]. Among these, physiological plant dysfunctions related to high temperatures and drought have been identified as the main causes of tree impairment and the emergence of *Botryosphaeriaceae*-related diseases [25,26]. Hence, climate change can be considered one of the major drivers of *Botryosphaeriaceae*-related attacks, and this has attracted the attention of researchers, especially in the Mediterranean region [27,28].

Some members of this family have been found in association with *Fraxinus* dieback and potentially involved in the etiology of the decline of ash formations [12,29–31]. In this study, we have investigated the causes of a widespread decline/dieback of *F. ornus* stands

in central and northern Italy (Tuscany, Veneto and Friuli Venezia Giulia), with identification of the fungal species involved, proof of pathogenicity and elucidation of the key role of some endophytic and canker-associated *Botryosphaeriaceae* in the development of the observed decline/dieback syndrome.

## 2. Materials and Methods

### 2.1. Field Surveys and Sampling

Field surveys were conducted in twenty-five *F. ornus* stands mixed with other broadleaved tree species, ranging in altitude from 50 to 722 m a.s.l., distributed in three Italian regions: Tuscany, Veneto and Friuli Venezia Giulia. Symptoms affecting *F. ornus* individuals were registered for a six-year period from 2018 to 2023 in each analyzed area and geographical coordinates were recorded for each stand (Table 1). In order to identify the causal agents, samples were collected from symptomatic shoots, branches and stems. Portions of stems with cankers were retrieved by dissecting the stems of 91 collected ash trees into 20 cm long fragments. Each collected sample was placed in a bag, classified by location and taken to the Plant Pathology Laboratory, DAGRI (University of Florence; Italy) for laboratory analyses.

**Table 1.** Sites surveyed for *Fraxinus ornus* dieback and number of collected samples.

| Sites | Region | Altitude (m. a.s.l.) | Geographic Coordinates | | Number of Samples Collected |
|---|---|---|---|---|---|
| | | | Latitude (X) | Longitude (Y) | |
| 1 | Tuscany | 498 | 43.065824° | 11.070266° | 3 |
| 2 | Tuscany | 448 | 43.055333° | 11.081245° | 6 |
| 3 | Tuscany | 439 | 43.050120° | 11.082984° | 4 |
| 4 | Tuscany | 722 | 43.033821° | 11.094413° | 3 |
| 5 | Tuscany | 554 | 43.024153° | 11.101669° | 1 |
| 6 | Tuscany | 553 | 43.023907° | 11.101867° | 3 |
| 7 | Tuscany | 545 | 43.021426° | 11.093438° | 3 |
| 8 | Tuscany | 312 | 43.014474° | 11.091555° | 3 |
| 9 | Tuscany | 306 | 43.002083° | 11.111248° | 4 |
| 10 | Tuscany | 323 | 43.091973° | 11.170991° | 4 |
| 11 | Tuscany | 334 | 43.090548° | 11.171011° | 4 |
| 12 | Tuscany | 347 | 43.142534° | 11.115229° | 5 |
| 13 | Tuscany | 333 | 43.140372° | 11.112651° | 5 |
| 14 | Tuscany | 431 | 43.172679° | 11.045179° | 2 |
| 15 | Tuscany | 438 | 43.165697° | 11.032799° | 2 |
| 16 | Tuscany | 609 | 43.153484° | 11.031911° | 4 |
| 17 | Tuscany | 556 | 43.154064° | 11.035591° | 4 |
| 18 | Veneto | 240 | 45.309950° | 11.773347° | 6 |
| 19 | Veneto | 216 | 45.289807° | 11.703996° | 5 |
| 20 | Veneto | 184 | 45.300556° | 11.767728° | 4 |
| 21 | Veneto | 259 | 45.318134° | 11.708450° | 2 |
| 22 | Veneto | 50 | 45.264882° | 11.717811° | 5 |
| 23 | Friuli Venezia Giulia | 567 | 45.988357° | 13.634403° | 3 |
| 24 | Friuli Venezia Giulia | 64 | 45.945518° | 13.552939° | 4 |
| 25 | Friuli Venezia Giulia | 187 | 45.8334681° | 13.580679° | 2 |

### 2.2. Fungal Isolations

Samples were first examined under a Leica Wild M8 stereoscope (Leica Microsystems, Heerbrugg, Switzerland) to assess the possible presence of pycnidia or ascomata. Fungal isolations were then conducted. Samples were surface-disinfected by

washing with 3% sodium hypochlorite solution (NaOCl) for 1 min and rinsing in sterile distilled water. After disinfection, plant tissues were dried on sterile absorbent paper. Outer bark was removed by using a sterile scalpel. Five fragments (4–5 mm$^2$) of inner bark and xylem tissues were then cut aseptically from the margin of the healthy and necrotic parts. Isolations were performed on 9 cm diameter plastic Petri dishes filled with 2% Potato Dextrose Agar (Liofilchem srl, Roseto degli Abruzzi, Teramo, Italy) amended with 250 mg L$^{-1}$ ampicillin + 10 mg L$^{-1}$ rifampicin. All Petri dishes were incubated in the dark at 24 ± 1 °C for 7 days. Subsequently, grown colonies were subcultured onto new 2% PDA Petri dishes.

### 2.3. Morphological Identification

Fungal cultures were initially grouped into morphotypes based on colony growth and phenotypic characteristics, including surface and reverse colony appearance observed after 10 days of incubation on PDA at 25 ± 1 °C in the dark, and morpho-biometric data of conidia. To induce the production of fruiting bodies and conidia by each fungal taxon, each isolated morphotype was transferred onto water-agar supplemented with autoclaved pine needles [32] and incubated at room temperature under UV light. After 14 days of incubation, each isolate was inspected under a Wild M8 stereoscope for the production of pycnidia. Using a sterile scalpel, pycnidia were collected and mounted on glass slides in 100% lactic acid and observed under a Zeiss light microscope (ZEISS, Jena, Germany). Morphology of conidia was determined at ×40 magnification by measuring 100 conidia from each morphotype (length × width). Based on morphological identification, the isolation frequency of *Botryosphaeriaceae* was calculated using the following formula: F = (NBot/NTot) × 100, where F is the frequency of *Botryosphaeriaceae*; NBot is the number of fragments from which *Botryosphaeriaceae* were isolated; and NTot is the total number of woody fragments from which fungi were isolated [33].

### 2.4. DNA-Based Identification

All fungal morphotypes were transferred onto sterile cellophane in 9 cm Petri dishes containing PDA and maintained in the dark at 24 ± 1 °C for 1–2 weeks. Approximately 80 mg of mycelium was harvested from the cellophane surface for each fungal morphotype and placed in a sterile 2 mL Eppendorf tube. DNA was extracted using a GenElute plant Genomic DNA Miniprep kit using standard protocol (Sigma Aldrich, St. Louis, MI, USA) and, finally, stored at −20 °C. Amplification of the Internal Transcribed Spacer ITS region (including spacers ITS1 and ITS2 and the internal 5.8S rDNA gene) was conducted using the universal primers ITS1/ITS4 designed by White et al. [34] and applying PCR mixture and conditions as described in Moricca et al. [35]. Amplification products were visualized by putting 5 µL of each PCR amplicon in a gel electrophoresis containing 1% agarose gel (Sigma-Aldrich), 1× Tris-acetate-EDTA (TAE) buffer and ethidium bromide (0.5 µg mL$^{-1}$) as staining. The approximate size (bp length) of the amplification products was determined by adding to the gel the 100 bp DNA ladder Ready to Load (Genespin). PCR amplicons were purified by using the Kit NucleoSpin® Gel and PCR Clean-up (Macherey-Nagel, Düren, Germany), following the manufacturer's instructions. Purified products were sent for forward and reverse sequencing to StarSEQ® GmbH (Mainz, Germany). The nucleotide sequences were read and edited with FinchTV version 1.4.0 (Geospiza, Inc.; Seattle; WA, USA, http://www.geospiza.com/finchtv, accessed on 3 October 2023). The identity of analyzed fungi was determined by comparing the obtained consensus sequences with that deposited on NCBI by applying the nucleotide BLAST (Basic Local Alignment Search Tool; http://www.ncbi.nlm.nih.gov/BLAST accessed on 28 September 2023) searches [36]. Generated sequences were submitted and deposited in NCBI GenBank (Table 2). Sequences were aligned with ClustalX v. 1.83 [37] using the following parameters: pairwise alignment parameters (gap opening = 10, gap extension = 0.1) and multiple alignment parameters (gap opening = 10, gap extension = 0.2, transition weight = 0.5, delay divergent sequences = 25%).

**Table 2.** Number of *Botryosphaeriaceae* species isolated from *F. ornus* at the investigated sites and GenBank accession numbers of one representative isolate per species.

| Species | No. of Isolates | Sites | ITS GenBank Code |
|---|---|---|---|
| *Botryosphaeria dothidea* | 53 [a] + 21 [b] | 1–20; 22–24 | OR119872 |
| *Diplodia fraxini* | 42 [a] + 15 [b] | 1–5; 9–22; 25 | OR177960 |
| *Diplodia subglobosa* | 3 [b] | 19; 24 | OR805517 |
| *Dothiorella iberica* | 2 [b] | 18 | OR805518 |
| *Dothiorella omnivora* | 2 [b] | 21 | OR805519 |
| *Neofusicoccum parvum* | 34 [a] + 3 [b] | 2; 4; 7–9; 11–15; 19; 23 | OR835588 |

[a] University of Florence collection. [b] University of Padua collection.

ITS sequences of representative isolates were edited and aligned in a dataset with 9 other sequences (ex-type sequences of 9 species belonging to 4 genera: *Botryosphaeria*, *Diplodia*, *Dothiorella* and *Neofusicoccum*) available in GenBank. Maximum likelihood (ML) analyses were performed with MEGA-X 10.1.8, including all gaps in the analyses. The best model of DNA sequence evolution was determined automatically by the software [38].

*2.5. Pathogenicity Tests*

Pathogenicity tests were conducted in an experimental field-plot at G.E.A. Green Economy and Agriculture, Centro per la Ricerca s.r.l., Pistoia, Tuscany (43.9192134° N, 10.9071857° E; http://www.cespevi.it, accessed on 1 June 2021) on 120 2-year-old seedlings of *F. ornus*. The experiment was performed from June to September 2021. Seedlings were potted using commercially produced loam and maintained in field conditions (outdoors) at 12 to 38 °C with drip irrigation in rows 30 cm apart. Inoculations were performed with three representative isolates of the most frequent fungal species (*B. dothidea*, *N. parvum* and *D. fraxini*) in the lower part of the stem, about 25 cm from the base of each seedling (about 1.2 cm in diameter). Thirty seedlings were used for each fungal species and 30 seedlings were mock-inoculated using a sterile PDA plug as control. The inoculation point was disinfected on each seedling with 90% ethanol, and a 6 mm diameter cork borer was used to remove the bark and expose the cambium. A 5 mm plug of mycelium of the tested fungal species, grown in Petri dishes in the dark on 2% PDA for 7 days at 25 °C, was inoculated in each seedling, with the mycelium side placed downwards into the wound. Mycelium was then covered with cotton soaked in sterile water and parafilm. To satisfy Koch's postulates, at the end of the trial, seedlings were taken to the laboratory, and re-isolations were performed from all the inoculated and control plants. Re-isolated fungal cultures were identified at species level by morphological and DNA-based analyses (ITS region amplification and sequencing). The pathogenicity of each tested fungal species was assessed by measuring the length of lesions (mm) developed on each seedling. Inspections, with lesion measurements, were conducted every two weeks.

*2.6. Statistical Analyses*

Pathogenicity test data were statistically analyzed by applying an analysis of variance (ANOVA). Significant differences between mean values were determined using Fisher's least significant differences (LSD) multiple range test ($p$ = 0.05) after one-way ANOVA using the software SPSS V.28 (IBM Corporate, Endicott, NY, USA). Differences achieving a confidence level of 95% were considered significant.

**3. Results**

*3.1. Field Surveys*

Symptoms were found on a total of 91 *F. ornus* plants and were visible on shoots, branches and stems. *F. ornus* plants were affected both in the adult and the juvenile stages, but the severity of the disease was greatest for the young regeneration, with high sapling mortality (Figure 1A–C). On adult trees, typical *Botryosphaeria* cankers were observed on

the main stem (Figure 1D–F). Cankers on the stem appeared as swollen lesions with bark cracks (Figure 1D,F), or as sunken, cracked bark areas (Figure 1E) presumably depending on the individual tree's readiness/capacity to respond to the infection with callus. When infection girdled the stem, the upper part of the crown died back.

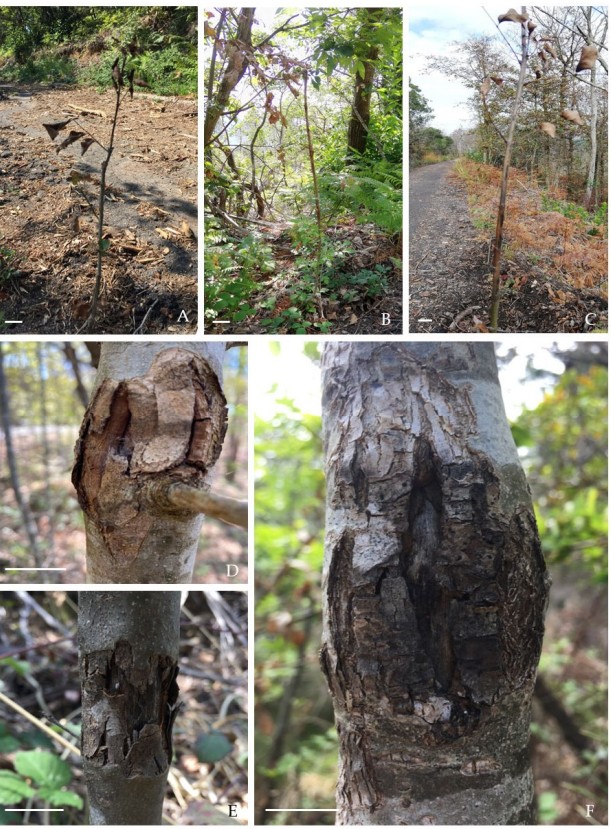

**Figure 1.** *Fraxinus ornus* individuals with symptoms of decline/dieback in the field (**A–F**); scale bar = 5 cm). Mortality of natural regeneration, with evident crown desiccation and cankers on the stem (**A–C**). A canker on the stem originating at a dead branch, with discolored bark and callus formation (**D**); sunken canker with cracked areas girdling the stem (**E**); a swollen canker with cracked areas on the bark (**F**).

Debarking of cankered parts of trees revealed the underlying necrotic lesions of the inner bark (secondary phloem) (Figure 2A–D).

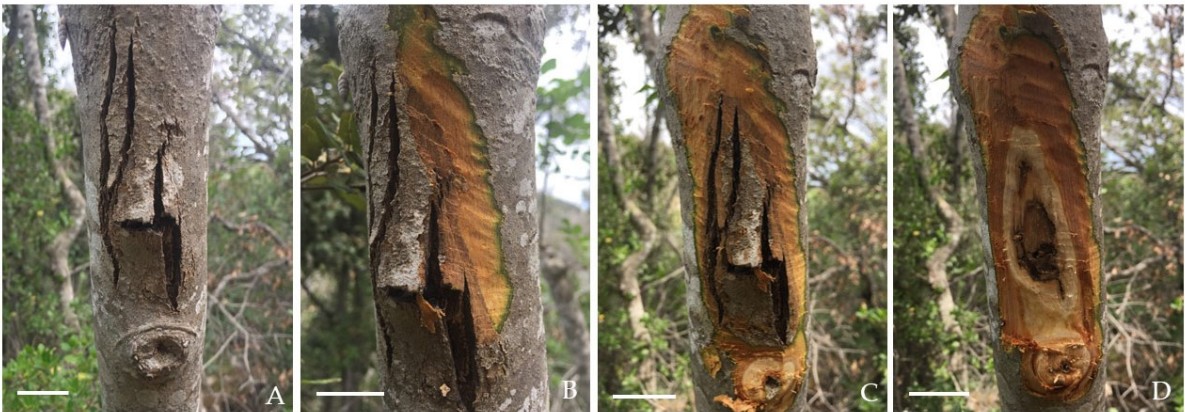

**Figure 2.** Visual representation in sequence of a longitudinal canker on a *Fraxinus ornus* trunk (**A**), subjected to progressive removal of the outer bark (**B–D**) to show the extension of the necrotic tissues throughout the inner bark (secondary phloem); scale bar = 5 cm.

The initial stages of the cankers were more easily observable on saplings, and they appeared as small, sunken, pale brown-purplish necroses (Figure 3A). Lesions then extended longitudinally, giving rise to narrow, elongated cankers that killed the sapling as soon as they girdled the stem (Figure 3B–D). More sporadically, cankers were observed on lateral branches of adult trees; as with the young regeneration, the distal part of the branch, beyond the canker, was dead (Figure 3E). On some saplings, cankers were diffused along the axis, revealing the ability of the causal agent to spread rapidly along the stem (Figure 3F,G). The multiple cankers caused dieback and death in most of the saplings and young trees.

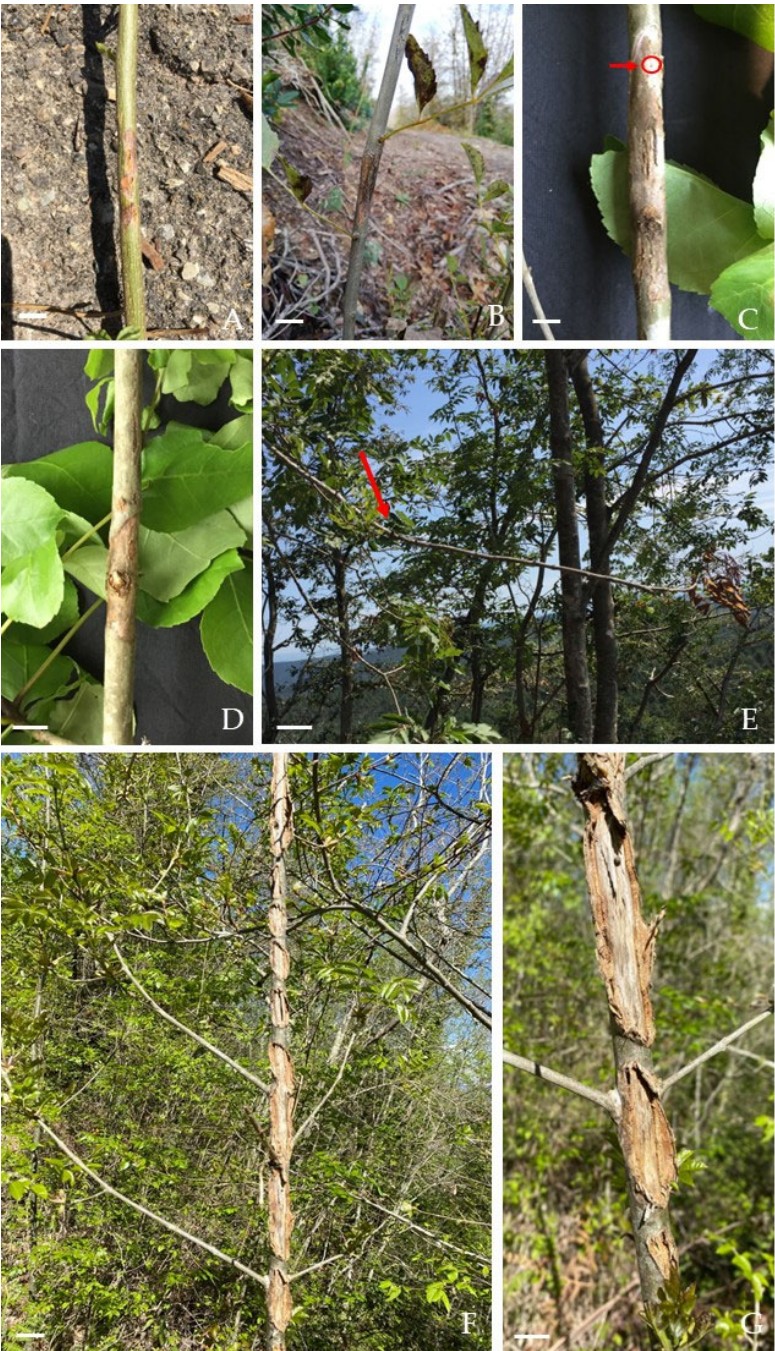

**Figure 3.** *Fraxinus ornus* individuals with symptoms of decline/dieback in the field (**A–G**). Incipient canker on a young sapling, with light brown to purplish-brown bark; individual lesions are coalescing to form a large necrotic patch (**A**), scale bar = 2 cm. More advanced, depressed canker, with beige-brown bark in advancing (outer) edge and dark-brown in its central portion (**B**), scale bar = 2

cm; Ellipsoidal canker with longitudinal cracks and occasional small black fruiting bodies (pycnidia) (circle) protruding through the bark (**C**), scale bar = 2 cm; Typical, sunken, light-brown canker with a marked edge is girdling the stem of an *F. ornus* sapling (**D**), scale bar = 2 cm; lateral branch of an adult *F. ornus*, desiccated in its distal portion due to a canker (arrow) (**E**), scale bar = 20 cm; young *F. ornus* tree with multiple cankers along the stem (**F**), scale bar = 4 cm; detail of cankers spread along the axis of a young *F. ornus* tree sapling seen at higher magnification (**G**), scale bar = 4 cm.

Leaf and shoot blight with crown dieback were also frequently observed (Figure 4A–E). *B. dothidea* was consistently isolated from cankers (more than 95%) on the stem of young and adult *F. ornus* individuals; *N. parvum* was isolated only sporadically (approximately 5% of cankers). *D. fraxini* was isolated at very high frequency (90%) from green, succulent shoot tissues and foliage, while approximately 10% of these tissues were infected with *N. parvum* (8%) or *B. dothidea* (2%).

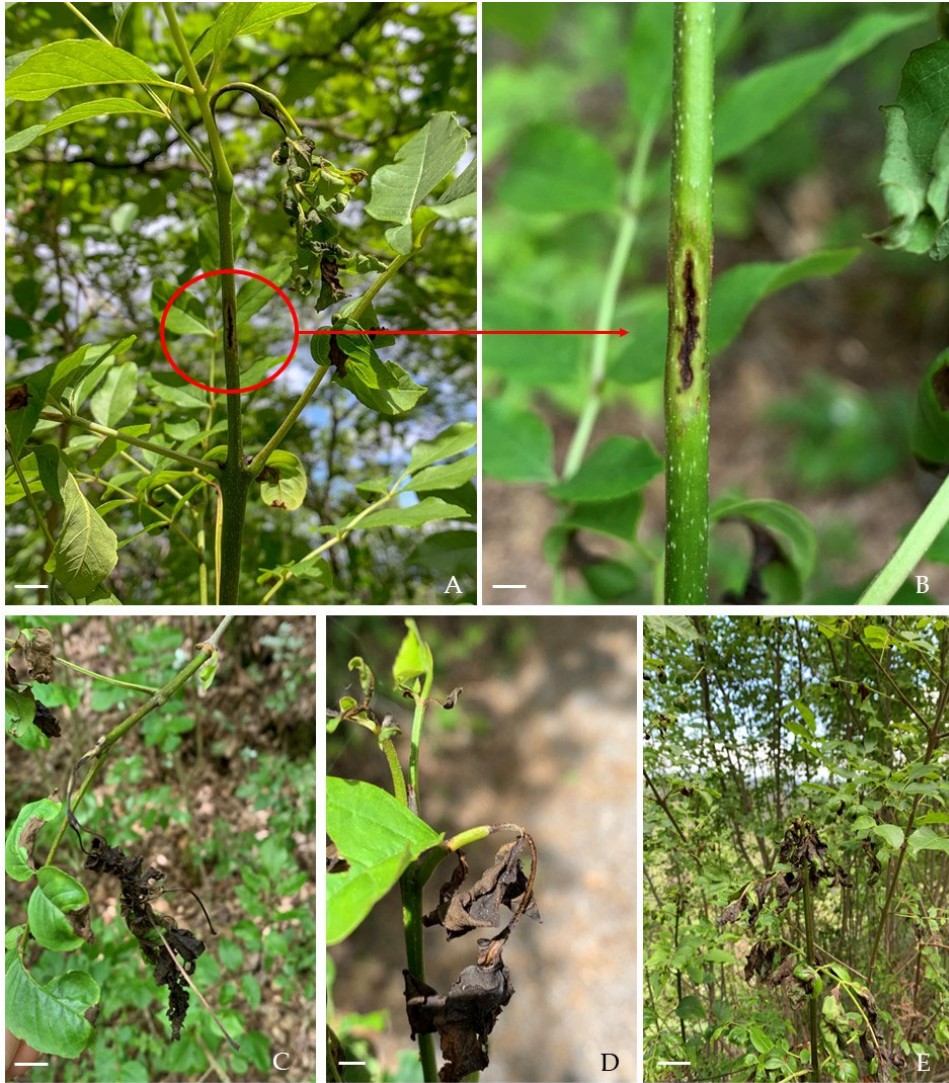

**Figure 4.** Symptoms on branches and shoots of young *F. ornus* in the surveyed areas (**A**–**E**). Shoot folding and initial stage of dieback of a young individual caused by *D. fraxini*; a longitudinal necrosis (red circle) is visible on the succulent, non-lignified stem (**A**), scale bar = 2 cm; higher magnification view (arrow) of the longitudinal, dark lesion on the stem of the individual in A (**B**), scale bar = 1 cm; dieback and necroses of shoots and tips on young *F. ornus* individuals (**C**–**E**), scale bar = 2 cm.

### 3.2. Isolate Identification

One hundred and sixty-five fungal colonies were identified based on morphological and molecular characteristics. The morphological traits of the isolates, including the

morphology and size of the conidia and shape of the colonies, were determined on MEA (Malt Extract Agar) and PDA, according to standard procedures. The colony of *B. dothidea* was grey or dark brown with a sparse aerial mycelium and a cottony surface texture. Conidia were fusiform or irregularly fusiform, base rounded, hyaline, and unicellular, rarely forming a septum. *D. fraxini* showed a white colony at first and dark grey later; conidia were hyaline, aseptate, smooth, and oblong to ovoid with a broadly rounded apex. The colony of *Neofusicoccum parvum* was initially white, becoming a greyish aerial mycelium typical of *Botryosphaeriaceae* species. Conidia were hyaline, unicellular, and ellipsoidal-shaped, with an obtuse apex and truncated base. *B. dothidea* was isolated from 23 out of the 25 sites, occurring in 74 of the 91 *F. ornus* plants investigated in total. *D. fraxini* was isolated from 57 of the analyzed *F. ornus* plants, while *N. parvum* was found in 37 of the total plants (Table 2). *B. dothidea* and *D. fraxini* pycnidia were also observed on branches with cankers by stereomicroscopic observations.

The other three species, *Diplodia subglobosa*, *Dothiorella iberica* and *Do. omnivora,* were isolated from a limited number of samples in only one or two sites (Table 2). DNA sequencing confirmed the identification of representative morphotypes as belonging to *B. dothidea* (OR119872), *D. fraxini* (OR177960), *D. subglobosa* (OR805517), *Do. iberica* (OR805518), *Do. omnivora* (OR805519) and *N. parvum* (OR835588) with BLAST searches that revealed complete (100%) homology of isolates with those of the above species deposited in the GenBank database. ITS-generated sequences were edited and aligned together with representative isolates of *B. dothidea*, *Neofusicoccum luteum*, *N. parvum*, *Diplodia corticola*, *D. mutila* and *D. fraxini*. The isolates clustered in well-supported clades (ML bootstrap > 95%) together with sequences of ex-type cultures (Figure 5).

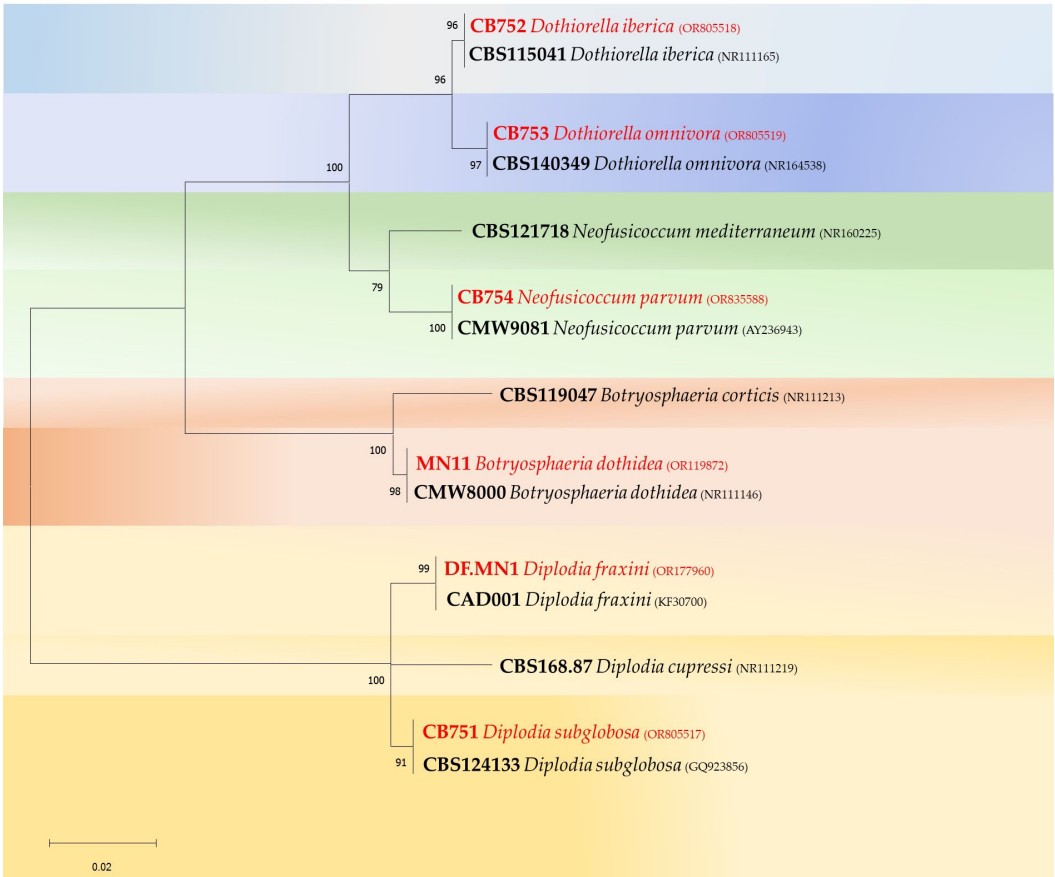

**Figure 5.** Maximum likelihood tree obtained from Internal Transcribed Spacer (ITS) sequences of *Botryosphaeriaceae* species. The tree is drawn to scale (0.01), with branch lengths measured in the number of substitutions per site. Bootstrap support values in percentage (1000 replicates) are given at the nodes. Ex-type cultures are in bold, and isolates obtained in this study in red.

### 3.3. Pathogenicity Tests

At the end of the pathogenicity trials, all *F. ornus* seedlings inoculated with *B. dothidea*, *D. fraxini* and *N. parvum* showed cankers and necrosis of different length around the infection court (Figure 6). By debarking cankered areas, dark brown inner lesions and internal discolorations were noticeable, corresponding precisely to the external lesions (Figure 6B). *B. dothidea* turned out to be the most aggressive pathogen, causing the larger cankers and under bark lesions (average length 58 ± 4.53 mm); *D. fraxini* caused lesions of 33 ± 1.72 mm; *Neofusicoccum parvum* caused less severe symptoms (mean lesions length = 22 ± 1.89) (Figure 7). With all the inoculated fungal species, lesion lengths were significantly greater than those recorded on control seedlings. Wounds appeared completely healed or with very small lesions at the mock-inoculation points on control seedlings (mean lesion length = 8 mm). The frequency of re-isolation of the three *Botryosphaeriaceae* species was 100%; no pathogenic fungal species were retrieved from *F. ornus* control seedlings. The identities of the reisolated species were confirmed by cultural, microscopic and molecular identification, thus fully fulfilling Koch's postulates. The pairwise comparison obtained from one-way ANOVA showed significant differences ($p < 0.05$) between the three species and the control concerning lesion lengths and internal discolorations.

Some of the infected seedlings, even those inoculated with the virulent *B. dothidea*, reacted vigorously to infection by forming a thick wound periderm around the infection court in order to promptly heal the wounds (Figure 6). Despite the vigorous reaction of the seedling, *B. dothidea* was able to spread rapidly upwards and downward along the axis—as already observed in the field—producing further cankers at a distance from the inoculation point (Figure 6A,B, arrows).

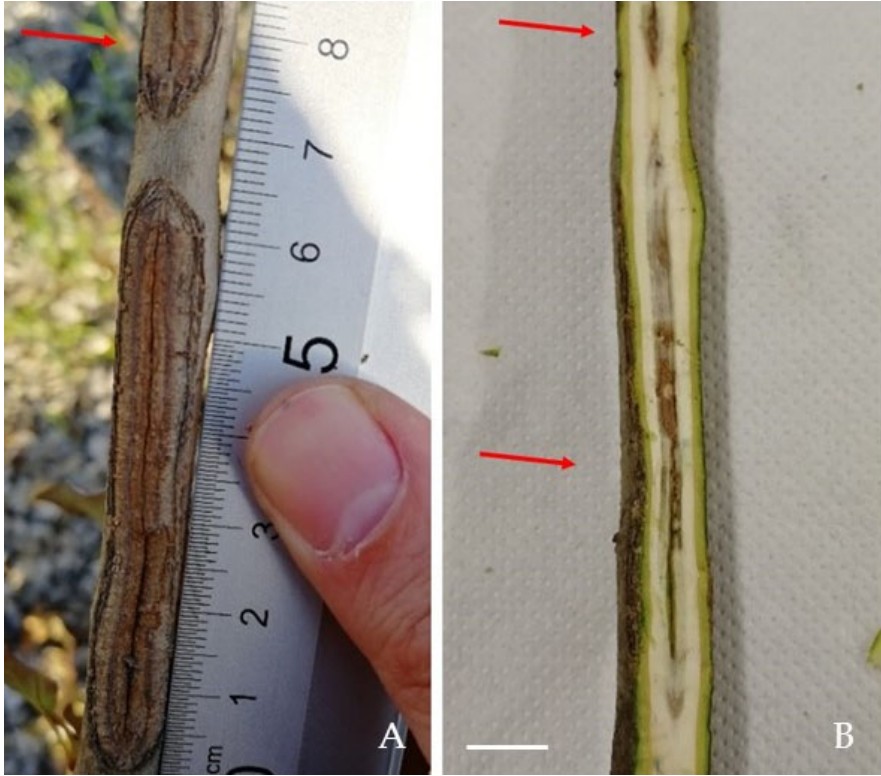

**Figure 6.** (**A**) Measurement of the extension of a canker artificially produced by inoculation with *B. dothidea* on an *F. ornus* seedling. A further canker, developed in the upper part of the image (arrow), proves the ability of the pathogen to spread rapidly along the stem, despite the seedling vigorously resisting infection, as is evident from the callus thickness. (**B**) Bark removal of cankered areas in the same seedling to observe the extent of the inner lesion highlighted the necrosis of the inner cambium, which was more pronounced in correspondence to the cankers (arrows), scale bar = 1 cm.

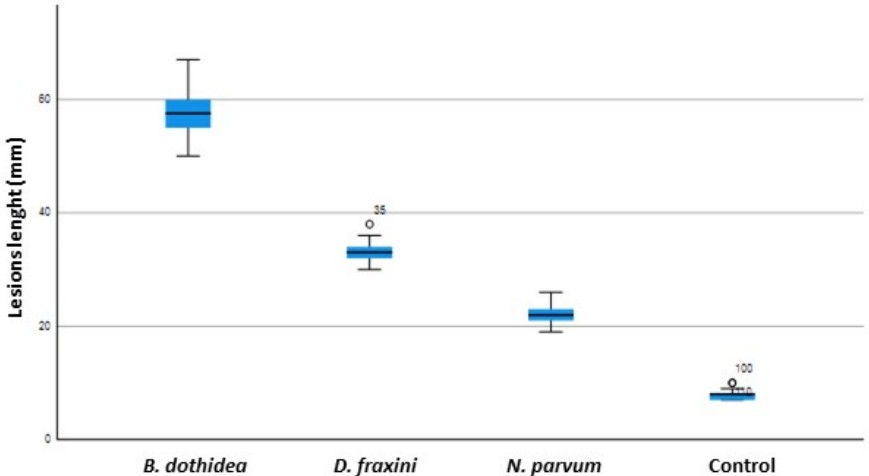

**Figure 7.** Graph depicting the extent of lesions produced on seedlings by artificial inoculation with *B. dothidea*, *D. fraxini* and *N. parvum*, compared to the control. Boxes represent the interquartile range, while the horizontal line within each box indicates the average value.

## 4. Discussion

The investigations conducted in this study provide compelling evidence that the dieback affecting *F. ornus* in forest formations of north-central Italy is to be ascribed to the action of prominent members of the *Botryosphaeriaceae* family. Symptoms similar to those caused by *Botryosphaeriaceae* on *F. ornus* have been reported on a plethora of woody hosts (fruit crops, tree plantations and natural forests) all over the world [22,23,29,39–41]. Although the *Botryosphaeriaceae* have been known as agents of diseases for some time, reports on their occurrence and deleterious impact on agricultural and forest systems have burgeoned in the last few decades [34,42,43]. This rise in *Botryosphaeriaceae* is probably due to multiple factors. First of all, there is the availability of more refined diagnostics tools enabling fungal taxa to be accurately identified, including cryptic and sibling species, also in those taxonomic groups, like the *Botryosphaeriaceae* family, with a taxonomic and nomenclatural history punctuated by confusion and controversy [22,23,44]. Global trade, especially the transnational movement of plant propagation material, has certainly played a key role in promoting the dispersal of some important members of this family into new, uncontaminated areas [45]. Anthropogenic disturbance, i.e., degradation fires, crop conversions, deforestation and inappropriate silvicultural interventions in general, by modifying agro-ecosystem processes and depleting host carbon reserves, make trees more vulnerable, thus more prone to infection and pervasive colonization by latent, opportunistic pathogens like the *Botryosphaeriaceae* [27]. However, most of the studies are today concordant in recognizing climate change as the major driver of phytopathological constraints induced by botryosphaeriaceous fungi [21]. Indeed, it is known that these fungi can live asymptomatically for an undefined period of time as endophytes inside plant tissues, switching to a pathogenic lifestyle when stressful environmental conditions, above all water stress, cause physiological impairment (e.g., reduced water transport across the apoplast) to their hosts [24–26,46]. Episodes of climate-related physiological damage to trees are becoming more and more frequent in the Mediterranean region (in which the area investigated in this study lies), which is considered today one of the most threatened by climate change. In fact, in the Mediterranean basin, the predicted global climate change scenario appears to be even more exacerbated, with warmer conditions, increased water deficits, heat waves and prolonged droughts being expected to increase over time [17–20]. The physiological stress caused to trees by climate vagaries and unusual weather events (included hail damage) is thus a precursor of massive tree infection by the *Botryosphaeriaceae* [47].

Studies dealing with *Fraxinus* dieback in Europe, including the Mediterranean areas, have mainly focused on assessing the pathogenic role of the ascomycete fungus *H. fraxineus* [4–7]. Only recently, this and a few other investigations have started taking into consideration the possible role of fungi other than *H. fraxineus*, among which the *Botryosphaeriaceae*, in the onset of ash decline/dieback [10,31,32]. Here, we provide evidence that six important fungi of the *Botryosphaeriaceae* family attack trees or parts of trees that are injured or in a weak or stressed condition. Three of these species in particular, namely *Botryosphaeria dothidea*, *Diplodia fraxini* and *Neofusicoccum parvum*, were consistently isolated from dead and dying *F. ornus* individuals. Among these, *B. dothidea* was isolated with the highest frequencies from stem wounds and was capable of producing the largest lesions in pathogenicity tests. *B. dothidea* grew in the living bark (phloem) and wood (xylem) and killed the branch or tree by girdling. The lower isolation frequency of the other two *Botryosphaeriaceae* species, taken together with the lower virulence they displayed in artificial inoculation tests, suggest that they are probably not primary agents of disease, although they are contributing to the overall decline of *F. ornus* trees. The high adaptation of *B. dothidea* is proven by its broad geographical distribution (it is a cosmopolitan fungus) and its wide host range, continuously updated and expanding [24,26]. Hence, adverse environmental conditions like those caused by climate change can favor this fungus in attacking and infecting hosts previously uninfected, creating a greater impact and potential expansion in different parts of the world [15,48–58]. *D. fraxini* has already been reported in studies on the etiology of diseases affecting *Fraxinus* [32]. Data on the current distribution of this fungus are still lacking also due to the recent taxonomic re-classification that has reassigned the name *D. fraxini* to this fungus, previously included in the *D. mutila* complex [10,59]. The adaptability of this fungus has been demonstrated by its reports from sites with different climatic conditions (average annual temperature from 6.1 to 15.1 °C and average annual precipitation ranging from 650 to 2050 mm), a fact that confirms the plasticity of this fungus against environmental changes [31]. *N. parvum* is a generalist, latent pathogen distributed worldwide, infecting a number of hosts in both fruit crops and native vegetation [60–64]. The widespread occurrence of this fungus reflects either its migration through the man-mediated movement of infected plant germplasm or its own ability to spread into wild or managed (e.g., plantations) ecosystems [41,65]. Although appearing more sporadic, the other three species are well known as pathogens involved in ash dieback [31]. *D. subglobosa*, in particular, plays a key role in the decline of common ash in north-eastern Italy and Slovenia [66]. As with *Diplodia fraxini*, this species was considered for a long time as belonging to the *Diplodia mutila* complex [31].

## 5. Conclusions

The serious problem of ash dieback by *H. fraxineus* has been misleading and has led to limitations in diagnostic and epidemiological investigations in some parts of Europe. This study adds essential knowledge to the etiology of *Fraxinus* decline/dieback and provides basic information on pathogenic fungi affecting *F. ornus* in the Mediterranean area. Here, fungi from the *Botryosphaeriaceae* family play a prevalent role in the development of the phenomenon, with *B. dothidea* and *D. fraxini* proving to be prevalent. These findings can have implications for future research and practice, better directing diagnostic efforts and monitoring campaigns.

**Author Contributions:** A.B., S.M. and B.T.L. conceptualization; A.B., S.M, C.B. and B.T.L. field survey, sample collection and assay; A.B., S.M., C.A., G.R. C.B. and B.T.L. data analysis; S.M. and B.T.L. funding acquisition; A.B. draft writing. All authors have read and agreed to the published version of the manuscript.

**Funding:** This research received no external funding.

**Data Availability Statement:** The sequence data presented in this study were deposited in the NCBI GenBank data repository.

**Acknowledgments:** This study was conducted by AB within her Ph.D. doctoral project at the University of Florence, Italy Ph.D. doctoral program in Agricultural and Environmental Sciences. We are grateful to the Unione dei Comuni della Val di Merse (SI) for the support during field surveys.

**Conflicts of Interest:** The authors declare no conflict of interest.

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
