# Peer review of "Botryosphaeriaceae Species Associated with Stem Canker, Shoot Blight and Dieback of Fraxinus ornus in Italy"

_forests, doi:10.3390/f15010051_

Round 1

Reviewer 1 Report

Comments and Suggestions for Authors

The manuscript describes the study on the aetiology of a novel Fraxinus ornus disease or syndrome. The problem is very well worked out with sufficient amount of data / observations to draw strongly supported general conclusions. In short, this is an interesting and valuable piece of work. The manuscript in generally is well written, only some minor corrections listed below:

Lines 73-75: This sentence can be understood as if the increase of temperature in the Mediterranean zone was expected to get higher than increases in other areas taken together. Please rephrase.

Line 100: altitude range stated here is 64 to 609 m.a.s.l. while in table 1 the lowest-ling location 20 is 50 m.a.s.l.

Lines 129-130: Manufacturer for the M8 stereomicroscope is provided for the second time. Is it required by the journal?

Line 130: I wanted to put some joke about East and West Germany, but I realized remembering this division makes me look old. Is this Zeiss microscope that old? Even if it is I would rather use just Germany.

Line 144: A DNA extraction kit does not recommend a protocol. Why not to say that DNA was extracted using a GenElute plant Genomic DNA Miniprepkit kit using standard protocol…

Lines 149-151: This sentence is cumbersome… one does not put amplicon in a gel electrophoresis. You would rather separate the PCR products in an agarose gel. Pleas rephrase.

Lines 163-165: The same for this sentence. Please rephrase for better reading and more clarity.

Figures 1, 2, 3, 4, and 6: There is a problem with A, B, C… marking of individual pictures. In the version I have only fig, 6B has its mark fully visible. For other pictures the marks are either half-visible (upper part) or not visible at all.

Lines 204-205: change to:

presumably depending on the individual tree's readiness/capacity to respond to the infection with callus

Line 210: change originated to originating

Line 211: change to: sunken canker with cracked areas girdling the stem

Lines 213-214: Change to: Debarking of cankered parts of trees revealed the underlying necrotic lesions of the inner bark (secondary phloem) (Fig. 2 A-D).

Figure 2: This is very good figure. Nice to see how this type of canker develops on various depths beneath the bark.

Line 219: Change to: Initial stages of cankers…

Line 224: diffused

Line 273: Change to: The isolates clustered in well-supported clades

Table 2: What PI and PD stand for? Put it in the table caption.

Line 309: remove “with a scalpel”

Lines 315-317: I don’t think this part is relevant here as this graph does not show any ANOVA results. Please remove.

Line 320: phytopathological framework?? Please rephrase.

Line 325: diseases

Line 326: impact on agriculture and forestry or impact on agricultural and forest systems

Line 368: change to updated and expanding

Lines376-377: from 6.1 to 15.1 °C of what? Annual average? Summer months’ average??...

Line 378: Change: “plasticity of this fungus to environmental changes as well” to: plasticity of this fungus against environmental changes

Lines 388-394: I would add here an information about the two most frequent species (B. dothidea and D. fraxini) as they were much more prevalent.

Good luck.

Comments on the Quality of English Language

The writing is generally OK, but some cumbersome sentences could be improved.

Author Response

REVIEWER 1

(in bold our responses)

Reviewer 1

The manuscript describes the study on the aetiology of a novel Fraxinus ornus disease or syndrome. The problem is very well worked out with sufficient amount of data / observations to draw strongly supported general conclusions. In short, this is an interesting and valuable piece of work. The manuscript in generally is well written, only some minor corrections listed below:

 Thank you for your appreciation of our work.

Lines 73-75: This sentence can be understood as if the increase of temperature in the Mediterranean zone was expected to get higher than increases in other areas taken together. Please rephrase.

There is copious scientific literature on the topic. The papers we have cited highlight precisely a worsening of climatic conditions (warming) in the Mediterranean area (widely recognized as a "hotspot for climate change").

Line 100: altitude range stated here is 64 to 609 m.a.s.l. while in table 1 the lowest-ling location 20 is 50 m.a.s.l.

Done. Thank you for reporting this oversight to us.

Lines 129-130: Manufacturer for the M8 stereomicroscope is provided for the second time. Is it required by the journal?

Done (removed).

Line 130: I wanted to put some joke about East and West Germany, but I realized remembering this division makes me look old. Is this Zeiss microscope that old? Even if it is I would rather use just Germany.

Done. We have left only “Germany”, as you suggested.

Line 144: A DNA extraction kit does not recommend a protocol. Why not to say that DNA was extracted using a GenElute plant Genomic DNA Miniprepkit kit using standard protocol…

Done.

Lines 149-151: This sentence is cumbersome… one does not put amplicon in a gel electrophoresis. You would rather separate the PCR products in an agarose gel. Pleas rephrase.

Done.

Lines 163-165: The same for this sentence. Please rephrase for better reading and more clarity.

Done. The sentence has been rephrased.

Figures 1, 2, 3, 4, and 6: There is a problem with A, B, C… marking of individual pictures. In the version I have only fig, 6B has its mark fully visible. For other pictures the marks are either half-visible (upper part) or not visible at all.

Probably there has been a formatting problem. Anyway, we’ve modified as you suggested.

Lines 204-205: change to:

presumably depending on the individual tree's readiness/capacity to respond to the infection with callus

Done.

Line 210: change originated to originating

Done.

Line 211: change to: sunken canker with cracked areas girdling the stem

Done.

Lines 213-214: Change to: Debarking of cankered parts of trees revealed the underlying necrotic lesions of the inner bark (secondary phloem) (Fig. 2 A-D).

Done.

Figure 2: This is very good figure. Nice to see how this type of canker develops on various depths beneath the bark.

Thank you.

Line 219: Change to: Initial stages of cankers…

Done.

Line 224: diffused

Done.

Line 273: Change to: The isolates clustered in well-supported clades

Done.

Table 2: What PI and PD stand for? Put it in the table caption.

Done.

Line 309: remove “with a scalpel”

Done.

Lines 315-317: I don’t think this part is relevant here as this graph does not show any ANOVA results. Please remove.

Done.

Line 320: phytopathological framework?? Please rephrase.

Done.

Line 325: diseases

Done.

Line 326: impact on agriculture and forestry or impact on agricultural and forest systems

Done.

Line 368: change to updated and expanding

Done.

Lines376-377: from 6.1 to 15.1 °C of what? Annual average? Summer months’ average??...

Done.

Line 378: Change: “plasticity of this fungus to environmental changes as well” to: plasticity of this fungus against environmental changes

Done.

Lines 388-394: I would add here an information about the two most frequent species (B. dothidea and D. fraxini) as they were much more prevalent.

Done.

Reviewer 2 Report

Comments and Suggestions for Authors

Peer-review report of the research article (forests-2778269)

The manuscript entitled, "Botryosphaeriaceae species associated with stem canker, shoot blight and dieback of Fraxinus ornus in Italy" is a good piece of research submitted for publication in the journal "Forests."

The authors have added a general introduction about the tree species, which requires more data regarding the morphological characteristics and abundance of the selected tree species. Moreover, the authors mentioned a bitter-sweet sap produced from the tree. What is the economic value of this sap? In which products it is used?

The authors need to add more facts about the selected pathogenic fungus's growth conditions and how it is possibly taking advantage of the changing environmental conditions.

The authors have mentioned the general importance of the Botryosphaeriaceae family regarding pathogenicity. It requires more facts, such as which members of this family cause important or prevalent diseases. Which tree or plant species do they target, and at what scale?

The authors have mentioned the findings related to the prevalence of different pathogenic species; however, more details are required. Is there any versatility among the species found from different infected portions of the plant, or are the same species found in all parts? What is the frequency of cankers found per plant? Is the prevalence of cankers random, or is a particular pattern followed?

What is the average length of the cankers found in wild species? Is there any difference between the characteristics of cankers on wild species and the cankers in the experimental area?

What is the depth of infestation of the pathogenic fungus? Is there any difference in depth among different pathogenic species?

Lines 299-304. The authors have mentioned the establishment of distant cankers. Is this phenomenon present in all pathogenic species? What is the frequency of such cankers? How are the distant cankers different from the initially produced cankers?

Are the leaves dying because of the fungal attack or the blockage of the nutrient transport system?

Did the authors observe any pathogenicity on branches and leaves during the field trial?

The authors must improve the images and make them more readable regarding the symptoms of the disease and indications of particular features by adding scale, etc.

The discussion seems generic and needs improvement regarding fact-based and specific comparisons regarding the pathogenicity of close relatives of the selected fungal species.

Comments on the Quality of English Language

Minor editing of the English language and grammar is required.

Author Response

REVIEWER 2

(in bold our responses)

 Moreover, the authors mentioned a bitter-sweet sap produced from the tree. What is the economic value of this sap? In which products it is used?

Done. We did not deem it necessary to dwell further so as not to divert attention from the key points of the manuscript, which concern phytopathological issues.

The authors need to add more facts about the selected pathogenic fungus's growth conditions and how it is possibly taking advantage of the changing environmental conditions.

Thanks for this suggestion, but the biology of the main pathogens isolated in this study is widely reported in the literature (more details are reported in https://doi.org/10.1007/s13225-014-0282-9 and https://doi.org/10.1111/mpp.12495), at the same time it is not among the objectives of this study.

The authors have mentioned the general importance of the Botryosphaeriaceae family regarding pathogenicity. It requires more facts, such as which members of this family cause important or prevalent diseases. Which tree or plant species do they target, and at what scale?

Botryosphaeriaceae is one of the most investigated families of pathogenic fungi worldwide (a search on SCOPUS using the terms “Botryosphaeriaceae and diseases” returns a list of 500 records starting from 2004). Recent interest in this family has been linked to the abilities to survive as endophytes and to change to pathogenic behaviour when host plants are under stress conditions. Many species belonging to Botryosphaeriaceae (e.g Diplodia corticola, Lasiodiplodia theobromae and Neofusicoccum parvum) can cause severe diseases of woody plants in natural and urban areas, nurseries and in agricultural crops. more details are reported in: https://doi.org/10.3114/sim0021. Further information on pathogenicity, ecology and impact of this group of fungi can be found in citation no. 24.

The authors have mentioned the findings related to the prevalence of different pathogenic species; however, more details are required. Is there any versatility among the species found from different infected portions of the plant, or are the same species found in all parts? What is the frequency of cankers found per plant? Is the prevalence of cankers random, or is a particular pattern followed?

The answers to some of these questions can be found in the website and bibliographical references indicated in our previous answers. Regarding the frequency of isolation of the different fungi and the portions of the tree from which they were isolated, these were reported from line 237 to line 242. The frequency of cankers per plant was not a parameter taken into consideration because in some cases (in plants of smaller diameter) a single canker was sufficient to kill the seedling, while in larger stems the cankers repeated along the axis indefinitely (see Figures 3F and 3G).

What is the average length of the cankers found in wild species? Is there any difference between the characteristics of cankers on wild species and the cankers in the experimental area?

Cankers in the wild varied from a few (2-3) cm to over 10 cm. Lengths of cankers developed in field inoculation trials are reported at lines 287 -290.

What is the depth of infestation of the pathogenic fungus? Is there any difference in depth among different pathogenic species?

They are bark canker agents, so they do not normally delve deeper beyond the cambium.

Lines 299-304. The authors have mentioned the establishment of distant cankers. Is this phenomenon present in all pathogenic species? What is the frequency of such cankers?

This phenomenon occurs mainly with B. dothidea, sporadically with N. parvum and D. fraxini. The isolation frequency was reported in the text.

How are the distant cankers different from the initially produced cankers?

No difference was observed.

Are the leaves dying because of the fungal attack or the blockage of the nutrient transport system?

Leaves die as a consequence of shoot  dieback (collapse of the transport system and tissue necrosis).

Did the authors observe any pathogenicity on branches and leaves during the field trial?

No, symptoms were almost entirely on the stem. In a very few instances foliar necroses were observed.

The authors must improve the images and make them more readable regarding the symptoms of the disease and indications of particular features by adding scale, etc.

The quality of images was found excellent (by anonymous referees). We don't have photos with better resolution. Furthermore, attemps to improve figure quality at the pc (e.g. by using a photo-editing program) makes the situation worse because other details of the image are lost.

Scale has been added to each image.

The discussion seems generic and needs improvement regarding fact-based and specific comparisons regarding the pathogenicity of close relatives of the selected fungal species.

To our opinion, the discussion is exhaustive enough and it highlights the key aspects. We have also reformulated the paragraph Conclusions" to highlight possible future developments of research, also in practical terms.

Reviewer 3 Report

Comments and Suggestions for Authors

This paper investigated six Botryosphaeriaceae species associated with stem canker, shoot blight and dieback of Fraxinus ornus in Italy. The topic is interesting and the authors define the problem in a scientifically rigorous manner and by designing a convincing solution approach.

The manuscript refers to related studies including the most recent ones in a proper way and derives the limitations and gaps in the pertinent literature therefore justifying the reason to carry out this particular study.

In the conclusion section, instead of summarizing what has been done so far, the authors might better underscore the scientific value added of their paper, and/or the applicability of their findings/results (i.e. Implications for research and practice). This section should also refer to limitations of the study and suggest potential future research (i.e. although the title of the section is 'conclusion and future work', there is no explicit ideas on what could be further research work on the subject).  

The paper is generally well written and well organized. It may be worthwhile to have this paper edited to improve the flow and readability of the manuscript.

Comments on the Quality of English Language

Minor editing of English language is required

Author Response

REVIEWER 3

(in bold our responses)

The manuscript refers to related studies including the most recent ones in a proper way and derives the limitations and gaps in the pertinent literature therefore justifying the reason to carry out this particular study.

Thank you for appreciating our work.

In the conclusion section, instead of summarizing what has been done so far, the authors might better underscore the scientific value added of their paper, and/or the applicability of their findings/results (i.e. Implications for research and practice). This section should also refer to limitations of the study and suggest potential future research (i.e. although the title of the section is 'conclusion and future work', there is no explicit ideas on what could be further research work on the subject).  

Done.

The paper is generally well written and well organized. It may be worthwhile to have this paper edited to improve the flow and readability of the manuscript.

Thank you.

Round 2

Reviewer 2 Report

Comments and Suggestions for Authors

The authors have improved the manuscript and addressed the questions and suggestions raised.